# Aging Promotes Spontaneous Liver Injury: Insights from Metabolic, Inflammatory, and Fibrotic Pathways in C57BL/6 Mice

**DOI:** 10.3390/biom15121727

**Published:** 2025-12-11

**Authors:** Poonam Sagar, Sathish Kumar Perumal, Ramachandran Rajamanickam, Ramesh Bellamkonda, Sundararajan Mahalingam, Natalia A. Osna, Karuna Rasineni, Kusum K. Kharbanda

**Affiliations:** 1Research Service, Veterans Affairs Nebraska-Western Iowa Health Care System, Omaha, NE 68105, USA; poonam181995@gmail.com (P.S.); sperumal@unmc.edu (S.K.P.); rrajamanickam@unmc.edu (R.R.);; 2Department of Internal Medicine, University of Nebraska Medical Center, Omaha, NE 68198, USA; 3Department of Biochemistry & Molecular Biology, University of Nebraska Medical Center, Omaha, NE 68198, USA; 4Department of Pharmacology and Experimental Neuroscience, University of Nebraska Medical Center, Omaha, NE 68198, USA; nosna@unmc.edu

**Keywords:** aging, liver injury, steatosis, hepatic fibrosis, inflammation, metabolic dysfunction

## Abstract

Aging is a critical factor influencing susceptibility to hepatic injury. In this study, the spontaneous development of liver injury with advancing age and potential sex-related differences in these processes are examined. This study focuses on key mechanisms such as fatty acid metabolism, immune response, and cellular stress in male and female C57BL/6 mice. Aged male and female mice (20 to 22 months old) exhibited higher body weight and an altered metabolic profile and fatty acid metabolism compared to their younger counterparts (8 to 10 weeks old). In addition, increased oxidative stress, cellular senescence, expression of inflammatory markers, and cytokines/chemokines levels were also observed in aged male and female mice compared to younger mice. Furthermore, the aged mice exhibited increased indices of hepatic fibrosis, evident from the upregulation of smooth muscle actin-α, collagen, and transforming growth factor-β. In conclusion, aging promotes spontaneous liver injury by increasing indices of oxidative stress, steatosis, inflammation, and fibrosis. These results highlight the impact of chronological age on the liver that can increase its susceptibility to secondary hepatic stressors such as alcohol, high-calorie diet, or hepatotropic infections. Understanding how metabolic and inflammatory pathways change with aging in males and females is essential for elucidating the mechanisms that drive chronic liver disease progression. These insights are particularly important for developing targeted, sex-specific prevention and therapeutic strategies for the aging population.

## 1. Introduction

Aging is a multifaceted process that influences various organs and biochemical processes throughout the body. With the increasing number of individuals aged 65 and older, this demographic shift has become a significant global concern. Current projections indicate that the aging population may reach 994 million by 2030 and could soar to 1.6 billion by 2050. These figures highlight the rapidly growing number of older adults and underscore the urgent need to address the various challenges associated with aging. It is becoming increasingly important to develop comprehensive healthcare strategies tailored to manage age-related conditions effectively, improve quality of life, and support healthy aging processes [1]. Achieving this goal requires a better understanding of the aging process and its impact on health.

Aging is associated with an increased risk of several non-communicable chronic disorders, including those affecting the liver [2,3]. The liver is integral to various bodily functions and engages in a wide range of physiological processes. These functions include metabolism and energy regulation, detoxification, bile production, synthesis of blood proteins and clotting factors, storage of vitamins and minerals, and immunological support [4]. While the liver demonstrates remarkable resilience during aging, hepatic cells do undergo cellular changes characteristic of aging that adversely affect their phenotype and function [5]. Particularly important are alterations in the sinusoids, which contribute to hepatocellular complications and significantly impact the health and quality of life of the elderly [6,7]. Associated with this age-related change, there is a progressive reduction in liver volume and weight, primarily due to a decline in the number of hepatocytes. This decline is accompanied by chromatin condensation and a slower DNA synthesis and repair process [8]. Furthermore, a significant drop in nicotinamide adenine dinucleotide (NAD+) levels has been observed in aged mice, mainly due to the downregulation of key regulators like proliferator-activated receptor gamma coactivator 1-alpha and sirtuin 1 (SIRT1). This reduction in NAD+ disrupts essential metabolic processes, leading to impaired lipid metabolism, glycolysis, and accumulation of triglycerides in the liver [9]. Aging hepatocytes experience cellular senescence, secreting pro-inflammatory cytokines and other mediators that contribute to age-related liver dysfunction [10,11,12,13]. Moreover, age-related loss of function in SIRT1 activates the NOD-, LRR-, and pyrin domain-containing protein 3 inflammatory pathway, leading to more severe hepatotoxin-induced liver injury [14].

Emerging evidence suggests different progress rates of aging among males and females as several sex-specific factors influence liver physiology and susceptibility to injury. A recent study reported that male mice exhibit a higher burden of aged cells throughout most of their lifespan compared to female mice. However, this sex-related difference diminishes with advanced age, at which point the proportion of senescent cells becomes comparable between males and females [15]. There is evidence demonstrating that female mice possess a greater stem-cell capacity for self-renewal, regeneration, and proliferation compared to male mice [16]. Another study highlights sexual dimorphism in aging with respect to the development of oxidative stress and inflammation [17]. Estrogen plays a key protective role in females by enhancing antioxidant defenses and mitigating oxidative damage. However, this protective capacity declines after menopause, leading to increased oxidative stress and consequently accelerating the rate of aging in post-menopausal females [17]. Thus, it would be interesting to see how the baseline characteristics of liver injury are influenced among different sexes.

These age-related and sex-dependent changes in the liver increase the susceptibility of older individuals to additional stressors, such as alcohol consumption, high caloric intake, and hepatotrophic viral infections. Particularly concerning is the impact of alcohol consumption on the elderly, given the rapid growth in episodic heavy drinking among this increasing population [1,18]. Our laboratory and others have shown that the earliest changes in the liver after alcohol consumption is the accumulation of fat in the liver that is associated with increases in oxidative stress, lysosomal, and proteasomal dysfunction [19,20,21,22,23,24]. With continued alcohol consumption, this early liver injury can progress to more severe liver damage characterized by inflammatory and cirrhotic changes [25,26,27,28].

Given our interest in understanding liver injury, this study was initiated to characterize baseline liver injury parameters in young and old mouse subjects with a particular focus on understanding how aging and sex differences influence hepatic vulnerability to injury. Since males and females display distinct metabolic and inflammatory profiles, we examined key markers of liver injury with a primary focus on those parameters relevant to our studies on alcohol-associated liver disease (ALD). Additionally, we examined various metabolic changes that occur with aging to gain a comprehensive understanding of aging-induced changes that could contribute to chronic liver diseases.

## 2. Methods

### 2.1. Animals

Young (8 to 10 weeks old) and aged (20 to 22 months old) C57BL/6 male (n = 5–10) and female (n = 5–10) mice were procured from Charles River laboratories. The animals were housed in Omaha Veterans Affairs Medical Center Veterinary Medical Unit accredited by the American Association for the Accreditation of Laboratory Animal Care (AAALAC) on a 12 h:12 h light/dark cycle with ambient temperature (20–25 °C) and 40–60% relative humidity with free access to a standard chow diet food and water. The study protocol was approved by the Institutional Animal care and Use Committee (IACUC) of Omaha Veterans Affairs Medical Center (IRBnet# 1729591, approved 27 January 2023). The animals were anesthetized with isoflurane (5% to effect), and the blood and liver were collected and appropriately processed for further analysis. The chest was opened, and death occurred from a combination of exsanguination and pneumothorax. These methods are consistent with the recommendations of the Panel on Euthanasia of the American Veterinary Association (AVMA Guidelines for the Euthanasia of Animals: 2020 Edition, AVMA, Illinois) and were approved by the Omaha VA IACUC.

### 2.2. Metabolic Behavior and Body Composition Analysis

Promethion metabolic mouse cages (Sable Systems, North Las Vegas, NV, USA) were used to house animals for metabolic screening and phenotyping. Animals were transferred from their home cages and singly housed in the metabolic cages. The animals were housed in metabolic cages for 72 h, with the first 24 h stated for the cage environment adaptation and the next 48 h for data collection. The horizontal and vertical movement of the mice across the three dimensions (X, Y, and Z) was measured using an XYZ IR Beam Break Activity Monitor. Food and water intake, body weight, VO_2_, and VCO_2_ were measured using Promethion precision MM-1 Load Cell Sensors. The time for metabolic parameters measurement was defined as a 12 h:12 h light/dark cycle. MetaScreen software (version 2.3.15.2) was used for multiplexed data acquisition and real-time monitoring. All metabolic phenotyping data were analyzed using ExpeData software with MacroInterpreter (version 2.50.6). Additionally, body fat composition was assessed using an Echo-MRI^®^ 3-in-1 whole body analyzer (EchoMRI, Houston, TX, USA).

### 2.3. Histopathology

Liver tissues were collected and fixed in 4% neutral buffered formalin. Fixed tissues were embedded in paraffin and sliced into 4 µm thick sections using a HistoCore BIOCUT R (Leica Biosystems, Dear Park, IL, USA). The sections were subjected to hematoxylin and eosin (H&E), and picrosirius red staining to assess liver histopathology and the intensity of fibrosis, respectively.

For H&E staining, liver sections were first deparaffinized in xylene and rehydrated through a graded alcohol series. The rehydrated sections were stained with hematoxylin, rinsed in water, and briefly differentiated in ethanol. Slides were then counterstained with eosin, followed by sequential dehydration in ethanol and clearing in xylene before final mounting [29]. For picrosirius red staining, liver sections were deparaffinized in xylene and rehydrated through a graded ethanol series. The slides were then incubated in Picrosirius Red solution (SSC1216-250, Cancer Diagnostics Inc., Durham, NC, USA) according to the manufacturer’s instructions. Following staining, sections were washed with 0.5% acetic acid, briefly dehydrated in absolute ethanol, cleared in xylene, and mounted for imaging [30]. The images were captured using a Keyence BZ-X810 microscope (Itasca, IL, USA) at various magnifications.

### 2.4. Triglyceride Quantification

Lipid extraction from liver homogenates was performed using the Folch method [31], as described previously [21,32]. Briefly, liver tissues were homogenized in a chloroform/methanol (2:1, *v*/*v*) mixture, followed by phase separation with the addition of 0.04% calcium chloride. After centrifugation, the upper aqueous phase was removed, and the lower chloroform phase containing the lipids was collected and dried using Savant SpeedVac SC110 Vacuum Concentrator (Holbrook, NY, USA). Total lipid extracts were saponified, and TGs were quantified using a triglyceride diagnostic kit (TR22421, Thermo Fisher Scientific, Waltham, WA, USA).

### 2.5. Lipid Peroxidation

Lipid peroxidation in liver homogenates was determined by measuring thiobarbituric acid reactive substances (TBARSs) as thiobarbituric acid (TBA) reacts with lipid peroxidation by-products, such as malondialdehyde (MDA) and 4-hydroxynonenal (4-HNE), as previously described [24]. Levels of MDA and 4-HNE were also determined by Western blot analyses.

### 2.6. Proteasomal and Lysosomal Activities

The chymotrypsin-like activity of the 20S and 26S proteasomes was measured using the SUC-LLVY-AMC fluorogenic substrate (G1100, UBPBio Inc., Dallas, TX, USA), following the protocol outlined in our previous publication [22,24]. The measurements were taken at excitation and emission wavelengths of 380 and 460 nm, respectively, using a fluorimeter (Spectramax M5, Molecular Devices, San Jose, CA, USA). Additionally, the trypsin-like activity of the 20S proteasome in liver tissue was assessed using the UBPBio diagnostic kit (G3100), as described in our earlier research [24]. Furthermore, lysosomal activities, including those of lysosomal acid lipase (LAL), cathepsin B, and cathepsin L, were evaluated using specific fluorogenic substrate kits, as detailed in our previous publications [24]. The catalytic activity of these hydrolases was determined in whole-cell lysates.

### 2.7. Gene Expression

The mRNA expression of selected genes was determined using real-time quantitative PCR (q-PCR). Total hepatic RNA was extracted using a PureLink RNA mini kit (12183025, Ambion, Austin, TX, USA), followed by treatment with DNase I to remove any DNA traces. The mRNA was converted to complementary DNA (cDNA) using a TaqMan Reverse Transcription reaction kit (3042979, Applied Biosystems, Foster City, CA, USA). q-PCR was performed on the Applied Biosystems 7500 q-PCR system using predesigned TaqMan^®^ FAM-labeled primers (4331182, Applied Biosystems) to assess changes in gene expression. Quantitative changes in the gene expression were determined by the ΔΔCT method relative to the housekeeping gene (β-actin) and represented as relative fold change. The primer details are listed in Appendix A. 

### 2.8. Protein Expression

Liver tissue was homogenized in chilled RIPA lysis buffer (pH 7.4), which contained 150 mM NaCl, 0.5% sodium deoxycholate, 0.1% SDS, 25 mM Tris, and 1% NP-40, supplemented with protease inhibitor cocktail (P2714, Sigma, St. Louis, MO, USA). The homogenate was then centrifuged at 14,000 RPM for 10 min, and the supernatant was collected for further quantification and Western blotting. For Western blotting, protein samples were separated using SDS-PAGE and transferred to a 0.4 µm nitrocellulose membrane. The membrane was subsequently blocked with 5% fat-free dry milk. The blots were incubated with primary antibodies, followed by the appropriate secondary antibodies. Protein bands were visualized using ECL substrate (170-15060, Bio-Rad, Hercules, CA, USA) on a Chemidoc MP imaging system (Bio-Rad). The intensities of the resulting bands were quantified using Image J and protein expression was normalized to β-actin serving as the housekeeping control. The antibodies used in this study are listed in Appendix A.

### 2.9. Statistical Analysis

The experimental data was analyzed using MS Excel and GraphPad Prism (version 10.5.0). The results are presented as mean ± SEM. To determine the statistical significance between the two groups, we used a *t*-test and one-way analysis of variance (ANOVA). A *p*-value of less than 0.05 was considered statistically significant.

## 3. Results

The study investigated the impact of aging on hepatic injury in C57BL/6 mice using male and female subjects aged 8 to 10 weeks old versus 20 to 22 months old mice. Physiological, metabolic, and biochemical analyses revealed significant liver changes with advancing age.

### 3.1. Aging Leads to Changes in Body Weight, Lean Mass, and Fat Mass

As expected, there was a significant increase in body weight, lean mass, and fat mass in both male and female aged mice (Figure 1). There was on average a 23.1% and 26.4% increase (*p* < 0.001) in body weight in aged males and female mice, respectively, compared to their younger counterparts (Figure 1a,d). However, the percent weight increase with age did not differ significantly between males and females. Aging resulted in a moderate but significant ~29% increase in the lean mass of both males and females (Figure 1b,e) but there was over a 2.5-fold increase (*p* < 0.01) in the fat mass in older subjects of both sexes compared to their respective younger counterparts (Figure 1c,f).

In line with their naturally nocturnal behaviors, both young and aged mice exhibit a notable reduction in intake of food/water (Appendix A), energy consumption, and pedestrian meters (Appendix A) during the day, regardless of sex. Further, we observed that there was a similar intake of water and food by the young and aged male during the entire day. While the aged female mice also had comparable food intake to the young mice during the entire day, these mice displayed a marked increase in water consumption at night while showing a substantial decrease during the day compared to their young counterparts (Appendix A).

About metabolism (Appendix A), aged mice of both sexes demonstrated a significant increase in overall energy consumption during the night, but only the aged females showed higher expenditure during the day. The respiratory quotient (RQ) for male mice remains steady and unaffected by age during the entire day. In contrast, aged female mice show a lower RQ than their younger counterparts, both during the day and at night. Activity assessed by determining the sum of all ambulatory locomotion of 1 cm/second or above within the x,y,z beam-brake system revealed that young and aged males and females have comparable step counts at night. However, while young and aged females show no difference in the locomotion during the day, the aged males take significantly fewer steps during the daytime (Appendix A).

Indirect calorimetry was utilized to assess the metabolic rates of both young and aged mice through measurements of oxygen (O_2_) consumption and carbon dioxide (CO_2_) production. Both young and aged male and female mice exhibited a significant reduction in CO_2_ production during the entire day when adjusted for body weight. Aged females showed minimal differences in metabolic rate relative to their younger counterparts, while aged males had a markedly lower normalized metabolic rate than young males. This trend was similarly observed in the normalized CO_2_ production for both sexes, as illustrated in Appendix A. Additionally, a significant decrease in metabolic rate normalized to lean mass was noted in aged males during both daytime and nighttime. In contrast, aged females maintained a consistent metabolic rate at night compared to young females but experienced a significant increase during the day. The CO_2_ production normalized to average lean mass reflected these trends as well (Appendix A).

Interestingly, average metabolic water production did not differ between young and aged mice at night. However, young mice displayed reduced metabolic water production during the daytime, regardless of sex. Furthermore, both males and females demonstrated significantly less movement in three-dimensional space during the day compared to the night, as expected, with no changes between the young and aged mice of both sexes, as shown in Appendix A.

### 3.2. Aging Promotes Hepatic Steatosis and Lipid Peroxidation

Aging leads to changes in fat metabolism that contribute to hepatic steatosis. Compared to young mice, both male and female aged mice exhibited an increase in hepatic steatosis, as indicated by a greater number of lipid droplets observed in H&E stains of the aged male (Figure 2a) and female mice (Figure 3a). However, the severity of steatosis does not differ significantly between aged males and aged females. Biochemical analysis revealed an increase in triglyceride levels (54.51 ± 6.17) compared to younger mice (38.21 ± 3.90, *p* < 0.05; Figure 2b) among males. While the liver triglycerides were numerically higher in the aged female mice compared to young mice, this difference was not statistically significant (Figure 3b). Both male and female aged mice exhibited notable changes in the mRNA expression levels of key fatty acid metabolism genes across both sexes. Specifically, a substantial increase in the fatty acid synthase (FASN) gene expression with a 13.07 ± 4.34-fold change in aged males (Figure 2c(i)) and a 9.33 ± 2.75-fold change in aged females (Figure 3c(i)) compared to their respective younger counterparts. Moreover, the expression of patatin-like domain 2 (PNPLA2), which plays a critical role in triglyceride hydrolysis, declined significantly by ~51% and ~59% in aged males and females (Figure 2c(ii) and Figure 3c(ii)), respectively. In addition, the expression of patatin-like domain 3 (PNPLA3), a lipid droplet-associated protein implicated in promoting liver injury, was increased in the livers of both aged males (Figure 2c(iii)) and females (Figure 3c(iii)) compared to their respective younger counterparts. The aged mice of both sexes demonstrated a significant (*p* < 0.05) rise in lipid peroxidation, as assessed by the TBARS assay. Densitometric analyses (Figure 4c and Figure 5c) of the immunoblots (Figure 4b and Figure 5b) revealed an increase in the levels of the two aldehyde products of lipid peroxidation, MDA and 4-HNE. These results highlighted a critical aspect of how aging contributes to disrupted lipid metabolism and increased cellular oxidative stress.

### 3.3. Aging Modulates Lysosomal and Proteasomal Enzyme Activities

We further investigated the activity of lysosomal enzymes. An essential enzyme in lipid catabolism, LAL, was decreased by ~34% (*p* < 0.05) in aged males and ~58% (*p* < 0.001) in aged females (Figure 6a and Figure 7a) compared to their respective younger counterparts. A decline in the activity of LAL is likely an important factor contributing to the accumulation of triglycerides in hepatocytes [33]. With regard to the cathepsins, our results indicated that cathepsin B and L proteolytic activities of aged male mice were comparable to those of young males (Figure 6b,c). In contrast, aged female mice showed a notable decline in cathepsin B and L proteolytic activity by 47.2% and 42.4%, respectively, compared to their respective younger counterparts (Figure 7b,c).

Additionally, we analyzed the proteasomal chymotrypsin-like and trypsin-like activities among young and aged mice of both sexes. The chymotrypsin-like remains consistent between young and aged males (Figure 6d). However, there was an increase in trypsin-like activity in the aged males (Figure 6e). In contrast, aged females display a noteworthy increase (*p* < 0.05) in chymotrypsin-like activity (Figure 7d), yet a significant decline (*p* < 0.05) in trypsin-like activity compared to their younger counterparts (Figure 7e). These differences in the proteolytic activities of the enzymes reflect the sex-dependent variations in enzyme activities among aged populations.

### 3.4. Cellular Senescence and Age-Dependent Hepatic Inflammation

Senescence is a process of cellular aging resulting from metabolic modification and in an irreversible cell cycle arrest. Additionally, as the organism age, the buildup of lipids and their byproducts significantly promotes oxidative stress causing detrimental health impacts [34]. This accumulation not only disrupts normal bodily functions but also triggers both type I and type II inflammatory responses and the subsequent release of inflammatory cytokines and chemokines [35]. As has been reported before, we observed an age-dependent upregulation of tumor protein 53 (p53), and cyclin-dependent kinase Inhibitor (p21), as assessed by the densitometric analysis (Figure 8b and Figure 9b) of the immunoblots (Figure 8a and Figure 9a).

In association with increased senescence and a senescence-associated secretory phenotype, we found a ≥ 2.5-fold increase in liver mRNA levels of tumor necrosis factor-α (TNF-α) (Figure 10a and Figure 11a) and interleukin-1β (IL-1β) (Figure 10a,b and Figure 11a,b) in both aged males and females compared with their respective younger counterparts, suggesting a noteworthy activation of the immune system. Additionally, we analyzed the mRNA levels of C-C motif chemokine ligand-2 (CCL2), also known as monocyte chemotactic protein-1 (MCP-1), which is involved in the recruitment of monocytes, memory T-cells, and dendritic cells in immune responses [36]. Our findings revealed that mRNA levels of CCL2 were significantly higher in aged males (4.81 ± 2.08-fold; Figure 10c) and females (3.48 ± 0.95-fold; Figure 11c) compared to their respective younger counterparts. Furthermore, a significant increase in the mRNA levels of C-X-C motif ligand 2 (CXCL2), alternatively known as macrophage inflammatory protein-2 (MIP-2), which plays an important role in recruiting neutrophils to areas of inflammation, was observed in aged males (2.42 ± 0.59-fold, *p* < 0.05; Figure 10d), while the females only showed a mild non-significant 1.16 ± 0.40-fold increase (Figure 11d) compared to their respective younger counterparts.

Further confirmation of the protein levels of some of these inflammatory cytokines through immunoblotting provided strong evidence for an increased inflammatory response in older age group mice of both sexes compared to their respective younger counterparts (Figure 10e and Figure 11e). Densitometric analysis revealed a significant 2.29 ± 0.11-fold increase in TNF-α protein expression in aged males, while aged females exhibited a 1.72 ± 0.21-fold increase (*p* < 0.05) compared to their younger counterparts (Figure 10f(i) and Figure 11f(i)). Additionally, protein levels of IL-1β were significantly elevated by ~2-fold in both aged males and females (Figure 10f(ii) and Figure 11f(ii)). Further examination of inflammatory markers, cluster of differentiation 68 (CD68) and myeloperoxidase (MPO) indicated increased macrophage and neutrophil infiltration in the livers of aged mice of both sexes compared to their younger counterparts (Figure 10e,f(iii,iv) and Figure 11e,f(iii,iv)). Together, these data indicate a notable rise in immune response with aging.

### 3.5. Aged Mice Demonstrate Signs of Fibrosis

The older male and female mice showed indices of hepatic fibrosis compared to younger counterparts, as shown in Figure 12 and Figure 13. The examination of liver sections using picrosirius red staining revealed a significant presence of collagen (red) in the livers of aged mice of both sexes (Figure 12a and Figure 13a). These collagen bundles indicate the initiation of hepatic fibrosis, which was further validated by biochemical analysis and by assessing the gene and protein expression of fibrotic markers by RT-PCR and Western blotting. Hydroxyproline levels and mRNA levels of transforming growth factor-β (TGF-β) and type I collagen pro-α-1 (Col 1α1) showed no significant changes among young and aged males (Figure 12b–d). However, aged females displayed ~2-fold increases in hydroxyproline, TGF-β, and Col 1α1 compared to those of younger individuals (Figure 13b–d).

Smooth muscle actin-α (α-SMA), which is a marker of activated hepatic stellate cells (HSC) and synthesizes aberrant extracellular matrix proteins, was found to be significantly upregulated in both aged males (3.04 ± 0.24-fold, *p* < 0.001, Figure 12e,f(i)) and females (2.30 ± 0.25, *p* < 0.01; Figure 13e,f(i)). Additionally, the protein expression of type II collagen (Col II) increased by 3.12 ± 0.20-fold in aged males (Figure 12e,f(ii)) and 1.84 ± 0.30-fold in aged females (Figure 13e,f(ii)).

## 4. Discussion

Aging significantly influences liver physiology and increases susceptibility to liver injury. This study aimed to explore the spontaneous development of liver injury in aged C57BL/6 mice, with a focus on the sex-related factors, mechanisms underpinning fatty acid metabolism, immune response, and cellular stress. Our results demonstrate that aging promotes several adverse changes in liver health, including increased steatosis, inflammation, and fibrosis, thereby enhancing the vulnerability of the liver to additional insults.

Previous studies have reported a decrease in liver volume by ~20–40% with aging from a reduction in size and number of hepatocytes [37]. There is also a 30–40% decrease in blood flow with age, which impacts the liver’s ability to clear out toxins and regeneration, a remarkable capability of the liver to activate compensatory mechanisms after injury, ensuring the continuation of essential physiological functions [38]. However, as the organism ages, the regenerative capacity of the liver declines, causing several pathologies. Experimental models of hepatic injury have consistently demonstrated age-related differences in susceptibility towards hepatic disorders [39]. The current study aimed to determine the baseline for liver injury parameters in young and aged mice of both sexes. We observed that aging promotes hepatic injury, which cumulatively enhances the risk of developing severe liver injury in the elderly with any second hits, including alcohol.

The increase in body weight, lean mass, and fat mass in aged mice, both male and female, aligns with known age-related changes in body composition. This shift is accompanied by altered metabolic profiles, as evidenced by increased energy consumption and disrupted respiratory quotients, particularly in female mice. The lower energy expenditure among aged males could be linked with fewer step counts during the day. These metabolic changes may reflect a shift in substrate utilization strategies that could be a contributing factor in the observed lipid accumulation in the liver. These changes emphasize the complexity of age-related adaptations, particularly between sexes, and highlight the need for further research into the underlying mechanisms that govern these differences.

Aging also markedly influences hepatic enzyme activities, especially that of LAL, whose activity decreased significantly in aged mice of both sexes compared to their younger counterparts. These results indicate that impaired lysosomal lipid catabolism in the aged mice could be a contributing factor in the development of hepatic steatosis, irrespective of sex.

Evidence from histopathology corroborates these biochemical changes, showing increased hepatic steatosis in aged mice. Elevated triglycerides and enhanced lipid peroxidation, as indicated by TBARS levels, HNE, and MDA, highlight the oxidative stress burden in the aging liver of both male and female mice. Other studies have also noted the accumulation of highly oxidized lipids and proteins known as “lipofuscin” (also referred to as “brown atrophy”) in the liver of aging individuals [40]. This oxidative stress likely contributes to senescence that exacerbates cellular damage and inflammation, setting the stage for more severe liver disease [41,42,43]. Increases in lipid accumulation from enhanced expression of FASN, sterol regulatory element-binding transcription factor-1 (SREBF1), and 3-hydroxy-3-methyl-glutaryl-coenzyme A reductase (HMGCR) in aged mice compared to younger controls have been reported [44,45]. Our results show similar trends in terms of lipid accumulation and FASN upregulation among aged groups of both sexes.

PNPLA2 is a key enzyme involved in triglyceride hydrolysis, and its intrinsic hydrolase activity is enhanced or inhibited by its interaction with other lipid droplet-associated proteins [46]. In our study, we found that mRNA levels of PNPLA2 were reduced, whereas PNPLA3 levels were higher in the aged groups of both sexes. The decrease in PNPLA2 expression corresponded with the accumulation of triglycerides in the liver of aged mice. Given that PNPLA3 inhibits the lipolytic function of PNPLA2, our findings suggest that the age-related increase in PNPLA3 levels further compromises the lipid breakdown efficiency of the already reduced PNPLA2 levels.

Inflammatory responses also become more pronounced with aging. p53 is a tumor suppressor protein that plays a critical role in regulating the cell cycle and preventing cancer by inducing cell cycle arrest, apoptosis, or senescence in response to DNA damage or other cellular stress. Thus, an increase in p53 denotes a cellular response to stress, including DNA damage, oxidative stress, or oncogenic activation, while p21 is a downstream effector of the p53 tumor suppressor pathway. Elevated mRNA and protein levels of inflammatory cytokines such as TNF-α, IL-1β, and chemokines, including CCL2 and CXCL2 in association with elevated expression of CD68 and MPO, indicate an active inflammatory milieu in the aged liver. This chronic inflammation could drive the progression from simple steatosis to more severe liver injury, including fibrosis.

Reactive oxygen species (ROS) and pro-inflammatory cytokines/chemokines are continuously produced because of persistent hepatic damage, which activates both resident and invading immune cells. These inflammatory mediators stimulate hepatocyte mortality and sustain immune cell recruitment, so continuing a cycle of injury and inflammation. Chronic inflammation triggers the activation and differentiation of HSCs, which differentiate into matrix-producing myofibroblasts. Over time, the imbalance between ECM production and degradation leads to progressive fibrosis, distortion of hepatic architecture, and eventually cirrhosis [47,48]. Our findings on fibrosis provide important insights into age-related liver damage. Picrosirius red staining revealed increased collagen deposition in aged livers, and the upregulation of fibrogenic genes such as TGF-β and type I collagen in aged females highlights a gender-specific vulnerability to fibrotic changes. The increased expression of proteins like α-SMA and type II collagen in both sexes further underscores the role of activated hepatic stellate cells in the development of fibrosis.

There were certain parameters such as cathepsin and chymotrypsin-like proteasomal activities, hydroxyproline levels, TGFβ1, collagen 1α1 expression that were comparable between the young and the aged male mice. It is possible that males are more resistant, and changes in these parameters could become apparent at a much later age than is seen in the females.

These results collectively suggest that aging exacerbates liver injury through a multifaceted process involving disrupted lipid metabolism, heightened oxidative stress, chronic inflammation, and fibrogenesis. The sex-specific differences observed, particularly in the fibrogenic responses, warrant further investigation to gather mechanistic insights and hormonal influences, such that we can tailor sex-dependent prevention and treatment strategies effectively.

## 5. Conclusions

Aged mice of both sexes demonstrate higher indices of hepatic steatosis, inflammation, and fibrosis compared to their younger counterparts. Our findings highlight the complex interplay between metabolic dysfunction, oxidative stress, and inflammatory responses in promoting age-related liver injury. Notably, the study also reveals sex-specific patterns in age-associated alterations, emphasizing that males and females differ in their susceptibility to hepatic decline during aging. These insights are particularly relevant for addressing liver injury from further insults like high-fat diets, alcohol consumption, or infections.

## Figures and Tables

**Figure 1 biomolecules-15-01727-f001:**
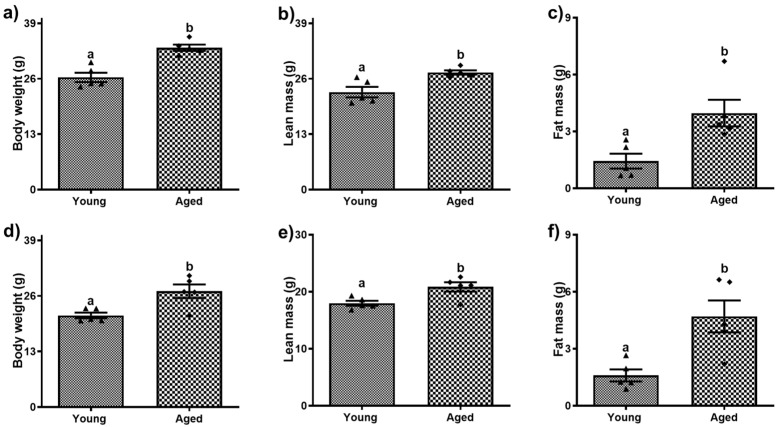
Body composition of young and aged male (**a**–**c**) and female (**d**,**e**) mice. (**a**,**d**) Body weight; (**b**,**e**) lean mass; (**c**,**f**) fat mass. Values are mean ± SEM (n = 5); values not sharing a common letter significantly differ from each other at *p* ≤ 0.05.

**Figure 2 biomolecules-15-01727-f002:**
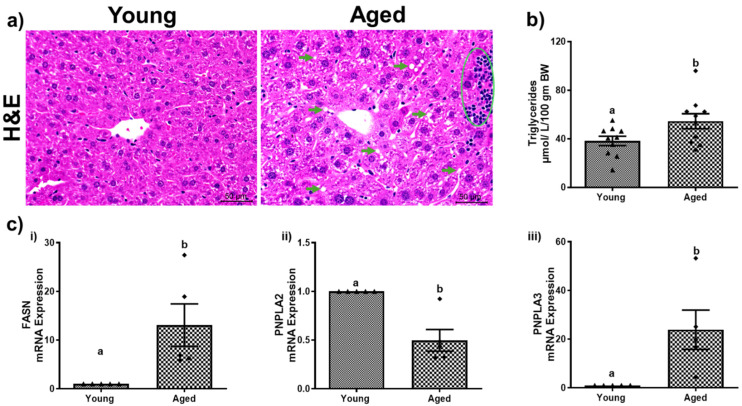
Aging promotes hepatic steatosis in male mice. (**a**) H&E-stained representative liver section (scale—50 µm) depicting numerous lipid droplets (green arrows) and inflammatory cell infiltrates (green circle); (**b**) hepatic triglycerides; (**c**) hepatic mRNA expressions of (**i**) fatty acid synthase (FASN); (**ii**) patatin-like phospholipase domain-containing protein 2 (PNPLA2), and (**iii**) patatin-like phospholipase domain-containing protein 3 (PNPLA3) of young and aged male mice. Values are mean ± SEM (n = 5–10); values not sharing a common letter significantly differ from each other at *p* ≤ 0.05.

**Figure 3 biomolecules-15-01727-f003:**
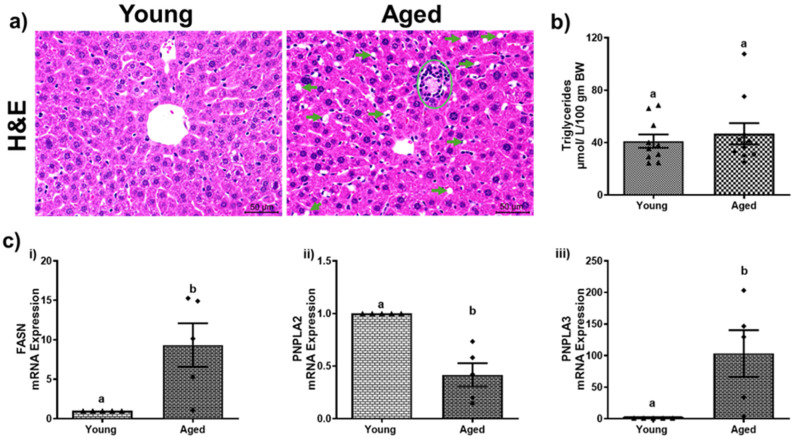
Aging promotes hepatic steatosis in female mice. (**a**) H&E-stained representative liver section (scale—50 µm) depicting numerous lipid droplets (green arrows) and inflammatory cell infiltrates (green circle); (**b**) hepatic triglycerides; (**c**) mRNA expressions of (**i**) fatty acid synthase (FASN); (**ii**) patatin-like phospholipase domain-containing protein 2 (PNPLA2), and (**iii**) patatin-like phospholipase domain-containing protein 3 (PNPLA3) of young and aged female mice. Values are mean ± SEM (n = 5–10); values not sharing a common letter significantly differ from each other at *p* ≤ 0.05.

**Figure 4 biomolecules-15-01727-f004:**
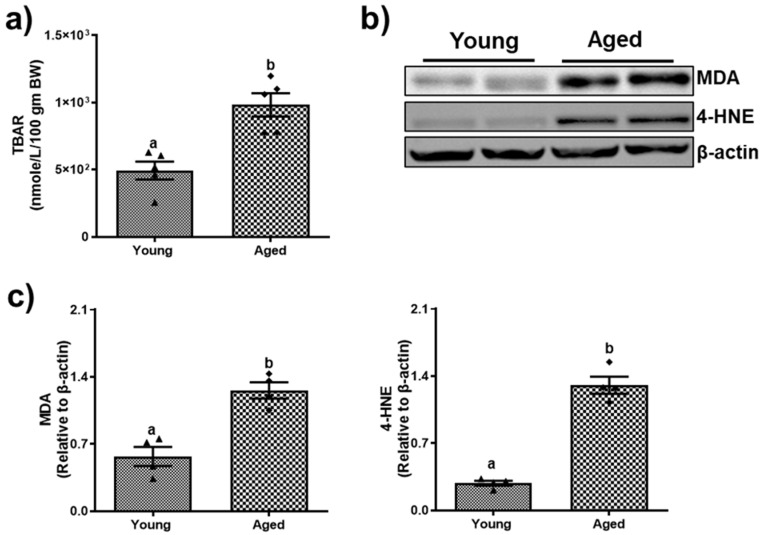
Aging promotes lipid peroxidation in male mice. (**a**) Hepatic thiobarbituric acid reactive substances (TBARS); (**b**) Western blots and (**c**) densitometric analysis illustrating hepatic malondialdehyde (MDA) and 4-hydroxynonenal (4-HNE) levels in young and aged male mice. Values are mean ± SEM (n = 5–10); values not sharing a common letter significantly differ from each other at *p* ≤ 0.05. Original images can be found in Appendix A.

**Figure 5 biomolecules-15-01727-f005:**
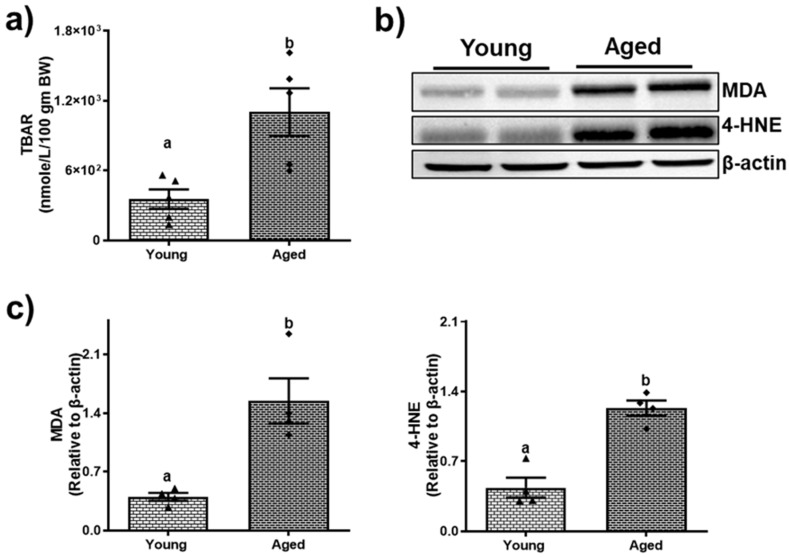
Aging promotes lipid peroxidation in female mice. (**a**) Hepatic thiobarbituric acid reactive substances (TBARS); (**b**) Western blots and (**c**) densitometric analysis illustrating hepatic malondialdehyde (MDA) and 4-hydroxynonenal (4-HNE) levels in young and aged female mice. Values are mean ± SEM (n = 5–10); values not sharing a common letter significantly differ from each other at *p* ≤ 0.05. Original images can be found in Appendix A.

**Figure 6 biomolecules-15-01727-f006:**
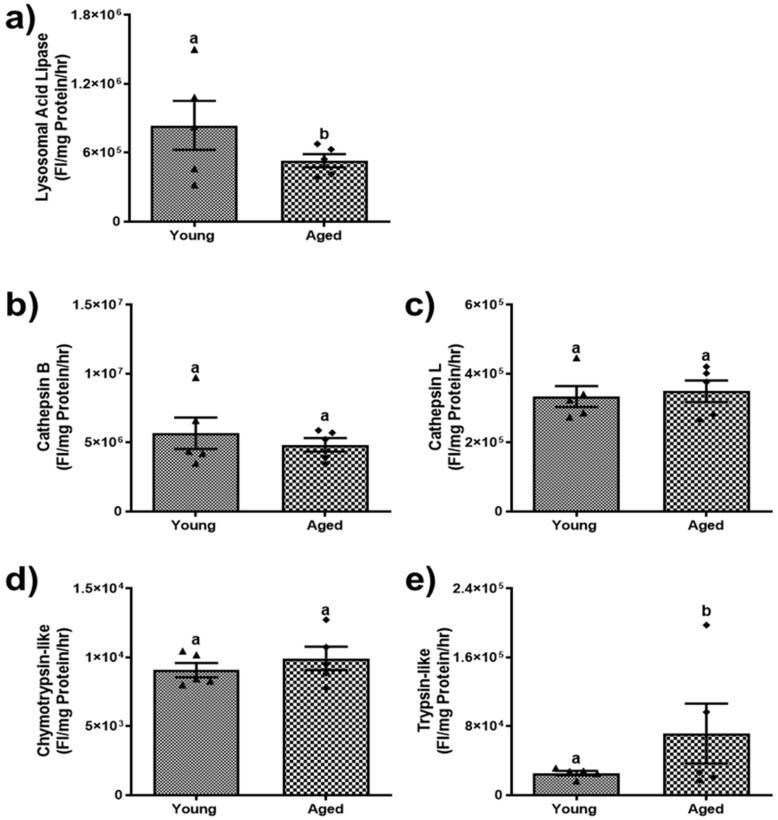
Lysosomal and proteasomal enzyme activities in livers of young and aged male mice. (**a**) Lysosomal acid lipase (LAL); (**b**) lysosomal cathepsin B; (**c**) lysosomal cathepsin L; (**d**) proteasomal chymotrypsin-like activity; (**e**) proteasomal trypsin-like activity. Values are mean ± SEM (n = 5–10); values not sharing a common letter significantly differ from each other at *p* ≤ 0.05.

**Figure 7 biomolecules-15-01727-f007:**
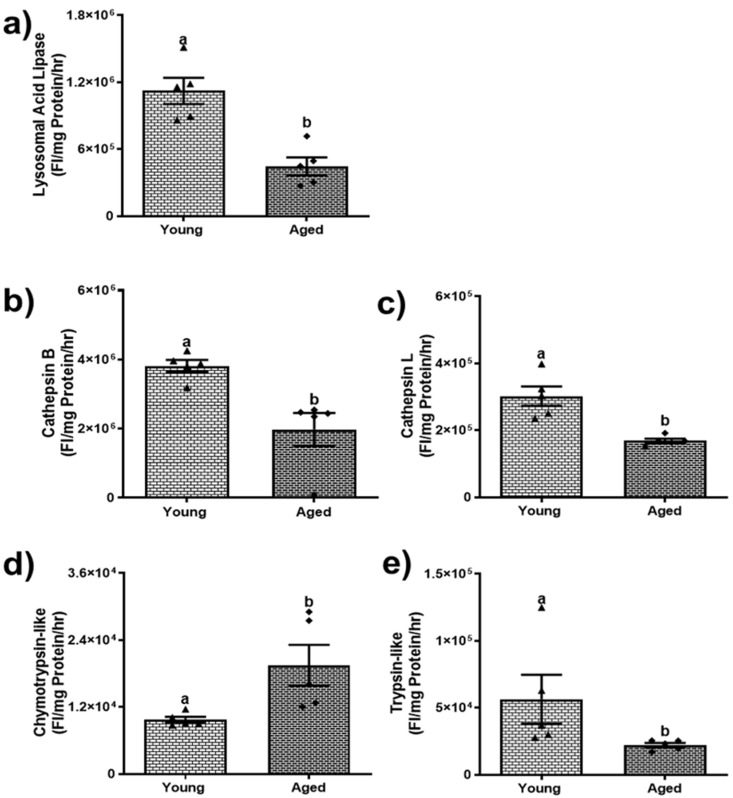
Lysosomal and proteasomal enzyme activities in livers of young and aged female mice. (**a**) Lysosomal acid lipase (LAL); (**b**) lysosomal cathepsin B; (**c**) lysosomal cathepsin L; (**d**) proteasomal chymotrypsin-like activity; (**e**) proteasomal trypsin-like activity. Values are mean ± SEM (n = 5–10); values not sharing a common letter significantly differ from each other at *p* ≤ 0.05.

**Figure 8 biomolecules-15-01727-f008:**
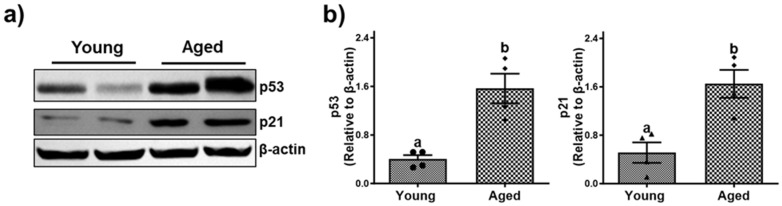
Aging-associated cellular senescence in livers of male mice. (**a**) Western blots and (**b**) densitometric analysis illustrating levels of the markers of senescence, cyclin-dependent kinase Inhibitor (p21), and tumor protein 53 (p53), and the loading control (β-actin) in representative liver lysates of young and aged male mice. Values are mean ± SEM (n = 5); values not sharing a common letter significantly differ from each other at *p* ≤ 0.05. Original images can be found in Appendix A.

**Figure 9 biomolecules-15-01727-f009:**
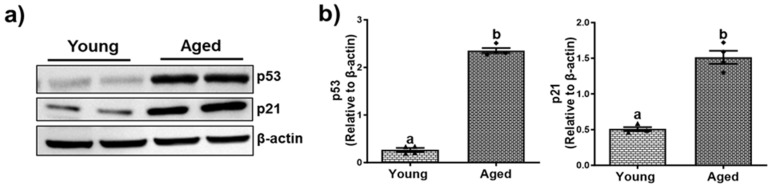
Aging-associated cellular senescence in livers of female mice. (**a**) Western blots and (**b**) densitometric analysis illustrating levels of the markers of senescence, cyclin-dependent kinase Inhibitor (p21), and tumor protein 53 (p53), and the loading control (β-actin) in representative liver lysates of young and aged female mice. Values are mean ± SEM (n = 5); values not sharing a common letter significantly differ from each other at *p* ≤ 0.05. Original images can be found in Appendix A.

**Figure 10 biomolecules-15-01727-f010:**
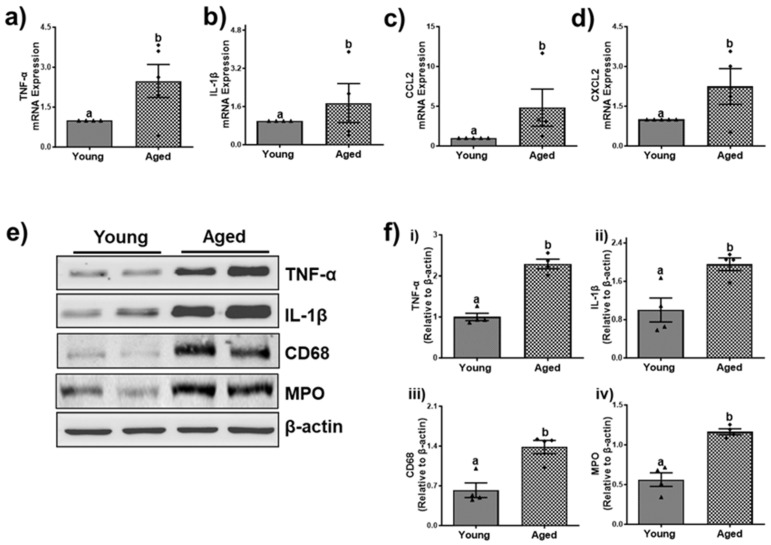
Aging promotes liver inflammation in male mice. Expression of mRNA encoding inflammatory cytokine and chemokines in hepatic tissue of young and aged male mice. (**a**) Tumor necrosis factor-α (TNF-α); (**b**) interleukin-1β (IL-1β); (**c**) C-C motif chemokine ligand-2 (CCL2); (**d**) C-X-C motif ligand 2 (CXCL2). (**e**) Western blots and (**f**) densitometric analysis illustrating levels of inflammatory cytokines (TNF-α, IL-1β,) and the markers of inflammation, cluster of differentiation 68 (CD68), and myeloperoxidase (MPO), reflecting macrophage and neutrophil infiltration. Values are mean ± SEM (n = 5); values not sharing a common letter significantly differ from each other at *p* ≤ 0.05. Original images can be found in Appendix A.

**Figure 11 biomolecules-15-01727-f011:**
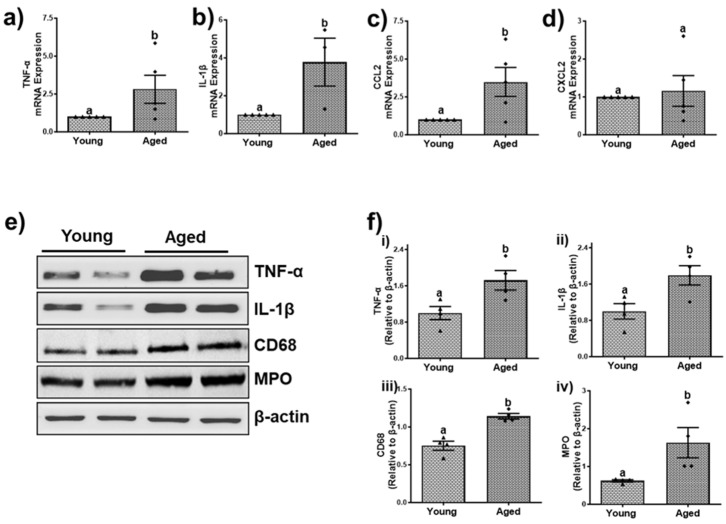
Aging promotes liver inflammation in female mice. Expression of mRNA encoding inflammatory cytokine and chemokines in hepatic tissue of young and aged male mice. (**a**) Tumor necrosis factor-α (TNF-α); (**b**) interleukin-1β (IL-1β); (**c**) C-C motif chemokine ligand-2 (CCL2); (**d**) C-X-C motif ligand 2 (CXCL2). (**e**) Western blots and (**f**) densitometric analysis illustrating levels of inflammatory cytokines (TNF-α, IL-1β,) and the markers of inflammation, cluster of differentiation 68 (CD68), and myeloperoxidase (MPO), reflecting macrophage and neutrophil infiltration. Values are mean ± SEM (n = 5); values not sharing a common letter significantly differ from each other at *p* ≤ 0.05. Original images can be found in Appendix A.

**Figure 12 biomolecules-15-01727-f012:**
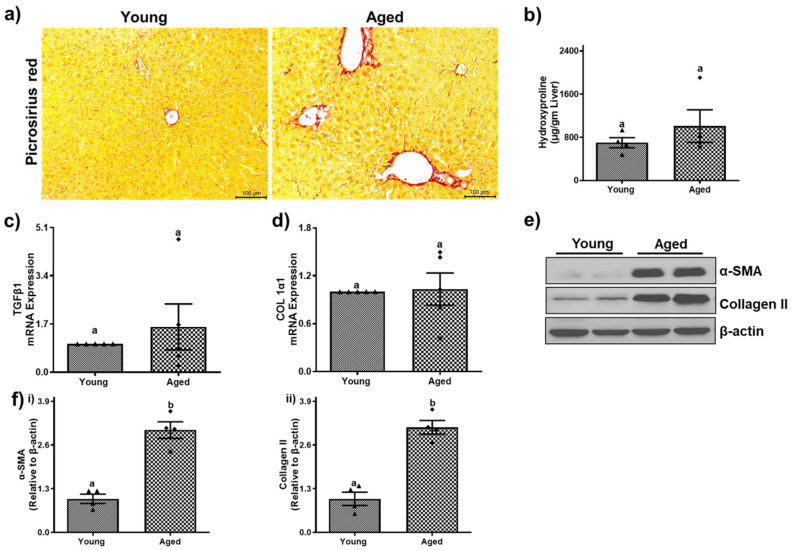
Aging promotes fibrosis development in male mice. (**a**) Picrosirius red staining showing collagen bundles (scale–100 µm). (**b**) Levels of hydroxyproline and (**c**) mRNA expression of (**c**) transforming growth factor β (TGF-β1), and (**d**) collagen (Col1α1). (**e**) Western blots and (**f**) densitometric analysis illustrating levels of fibrotic markers (α-smooth muscle actin (α-SMA), collagen II, and loading control (β-actin) in representative liver lysates of young and aged male mice. Values are mean ± SEM (n = 5); values not sharing a common letter significantly differ from each other at *p* ≤ 0.05. Original images can be found in Appendix A.

**Figure 13 biomolecules-15-01727-f013:**
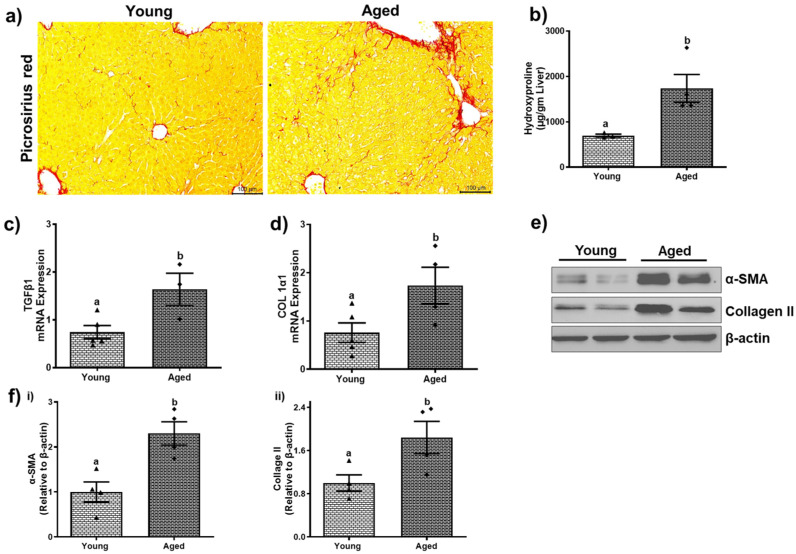
Aging promotes fibrosis development in female mice. (**a**) Picrosirius red staining showing collagen bundles (scale–100 µm). (**b**) Levels of hydroxyproline and (**c**) mRNA expression of (**c**) transforming growth factor β (TGF-β1) and (**d**) collagen (Col1α1). (**e**) Western blots and (**f**) densitometric analysis illustrating levels of fibrotic markers (α-smooth muscle actin (α-SMA), collagen II, and loading control (β-actin) in representative liver lysates of young and aged female mice. Values are mean ± SEM (n = 5); values not sharing a common letter significantly differ from each other at *p* ≤ 0.05. Original images can be found in Appendix A.

## Data Availability

The original contributions presented in this study are included in the article/Appendix A. Further enquiries can be directed to the corresponding authors.

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
