# Peer review of "Aging Promotes Spontaneous Liver Injury: Insights from Metabolic, Inflammatory, and Fibrotic Pathways in C57BL/6 Mice"

_biomolecules, 2025, doi:10.3390/biom15121727_

Round 1
Reviewer 1 Report
Comments and Suggestions for Authors
This original research is investigating the spontaneous development of liver injury with aging, focusing on mechanisms such as fatty acid metabolism, immune response, and cellular stress in male and female C57BL/6 mice. Since the manuscript is generally well prepared and the topic is relevant and potentially impactful, I recommend the manuscript for publication after minor revision. Detailed comments are provided below:
- The section in the introduction discussing alcohol consumption in the elderly is not relevant to the study and should be shorten.
- Please, check typographic mistakes (for example unnecessary spaces after citations “[6,7]”, before “[16]”, etc.).
- Additionally, in introduction section ensure that the abbreviation ALD is defined at its first appearance in the manuscript to maintain clarity for readers who may not be familiar with the term.
- The Introduction does not address what is currently known about sex-related differences in aging, despite the fact that both sexes were included in the study. Please add at least one paragraph summarizing existing evidence on how aging processes differ between males and females, particularly in relation to liver physiology and injury.
- The stated aims refer to alcohol effects, although the study does not include an alcohol model. Please revise the aim so it accurately reflects the true focus of the study.
- Also, the study aim does not mention that sex-related differences were evaluated, although both male and female animals were included in the analysis. Please revise the Aim section to clearly state that the study also investigates potential differences between sexes.
- In the section describing animal sacrifice (2.1), you state that the method is consistent with specific guidelines or recommendations. Please add an appropriate citation to support this statement.
- In the section 2.2. provide the references or manufacturer protocol numbers for the Sable Systems Promethion platform and Echo-MRI equipment. Additionaly, please describe how motion tracking in X/Y/Z dimensions was recorded. Clarify whether animals were single-housed during metabolic cage testing and whether this differed from baseline housing.
- In the section 2.3 Histopathology, the methodology for Hematoxylin & Eosin (H&E) and Picrosirius Red staining is described too briefly. These are critical techniques for your histological evaluation, so please explain specific procedures during staining. Also, adding appropriate citations will strengthen the reliability and reproducibility of these methods. How did you quantify fibrosis or steatosis? Please discuss in more detail in this section.
- In the section 2.4. describe in detail how did you created liver homogenates and add a citation. You explained liver homogenization in the section 2.8., but it should be discussed when first mentioned.
- In the section 2.7., please provide a list of all primer sequences (gene, forward and reverse primary) used in the RT-qPCR analysis
- For clearer visualization, it would be preferable to present the results for both sexes on the same graphs rather than in separate. This would allow readers to more easily identify sex-specific differences. In addition, please include the appropriate statistical comparison between males and females to demonstrate whether these differences are significant.
- Several sections of the Results contain interpretive or speculative statements, as well as background explanations regarding senescence and the roles of p53/p21. These elements should be moved to the Discussion section to maintain a clear separation between data presentation and interpretation. Please revise accordingly.
- Additionally, it should be clarified whether lipid droplet quantification was performed or if the conclusions are based solely on representative histological images. The same applies to the picrosirius red staining, therefore please specify whether collagen content was quantified (e.g., percentage collagen area) or if the assessment is purely qualitative.
- The Discussion should elaborate more thoroughly on the sex-related differences observed in your results. Please relate these findings to existing literature on sex-specific aspects of aging and liver injury to better contextualize the significance of your data.
- Conclusion also should give special focus on sex-related differences.
- Rewrite abstract with revised conclusion and aim based on sex-related differences.
Author Response
We thank reviewers for their helpful comments on the revision of the manuscript. In this revised version, we have addressed all the comments of the reviewer. Additions in the revised manuscript are highlighted.
Please see the attachment.

Reviewer 2 Report
Comments and Suggestions for Authors
The article is very interesting and useful for researchers and clinicians. A deeper understanding of the mechanisms of aging is essential for future effective therapy.
Recommendations:
- The authors should comment on the following aspect: is it possible that the pathological phenomena described during senescence would be different if the animals were free and had to procure their own food and water, which implies greater physical effort, compared to those in the study with free access to a standard chow diet food and water?
- Did the authors observe any correlation between abdominal adiposity and the severity of hepatic steatosis?
- The authors should clarify that the hepatic changes observed in the aging pattern are consistent with those found in non-alcoholic fatty liver disease.
- The authors should comment on the pathogenetic mechanisms involved in the connection between chronic liver inflammation and fibrosis.
- The authors could mention in the Discussion chapter if they intend to study different diets or drugs in the future that could mitigate liver injury during senescence through the reduction of lipid metabolism disorders, oxidative stress, chronic inflammation, or fibrogenesis.
Author Response
We thank reviewers for their helpful comments on the revision of the manuscript. In this revised version, we have addressed all the comments of the reviewer. Additions in the revised manuscript are highlighted.
Please see the attachment
